# CD163 Monoclonal Antibody Modified Polymer Prodrug Nanoparticles for Targeting Tumor-Associated Macrophages (TAMs) to Enhance Anti-Tumor Effects

**DOI:** 10.3390/pharmaceutics15041241

**Published:** 2023-04-14

**Authors:** Zun Yang, Haijiao Li, Wenrui Zhang, Mingzu Zhang, Jinlin He, Zepeng Yu, Xingwei Sun, Peihong Ni

**Affiliations:** 1State and Local Joint Engineering Laboratory for Novel Functional Polymeric Materials, Jiangsu Key Laboratory of Advanced Functional Polymer Design and Application, Suzhou Key Laboratory of Macromolecular Design and Precision Synthesis, College of Chemistry, Chemical Engineering and Materials Science, Soochow University, Suzhou 215123, China; 2Center for Medical Ultrasound, The Affiliated Suzhou Hospital of Nanjing Medical University, Suzhou 215001, China; 3Intervention Department, The Second Affiliated Hospital of Soochow University, Suzhou 215004, China

**Keywords:** anti-tumor, TAMs, CD163 monoclonal antibody, targeted drug delivery, prodrug, click reaction, doxorubicin, immunotherapy

## Abstract

Tumor-associated macrophages (TAMs)-based immunotherapy is a promising strategy. Since TAMs are mainly composed of M2-type macrophages, they have a promoting effect on tumor growth, invasion, and metastasis. M2-type macrophages contain a specific receptor CD163 on their surface, providing a prerequisite for active targeting to TAMs. In this study, we prepared CD163 monoclonal antibody modified doxorubicin-polymer prodrug nanoparticles (abbreviated as mAb-CD163-PDNPs) with pH responsiveness and targeted delivery. First, DOX was bonded with the aldehyde group of a copolymer by Schiff base reaction to form an amphiphilic polymer prodrug, which could self-assemble into nanoparticles in the aqueous solution. Then, mAb-CD163-PDNPs were generated through a “Click” reaction between the azide group on the surface of the prodrug nanoparticles and dibenzocyclocytyl-coupled CD163 monoclonal antibody (mAb-CD163-DBCO). The structure and assembly morphology of the prodrug and nanoparticles were characterized by ^1^H NMR, MALDI-TOF MS, FT-IR UV-vis spectroscopy, and dynamic light scattering (DLS). In vitro drug release behavior, cytotoxicity, and cell uptake were also investigated. The results show that the prodrug nanoparticles have regular morphology and stable structure, especially mAb-CD163-PDNPs, which can actively target TAMs at tumor sites, respond to the acidic environment in tumor cells, and release drugs. While depleting TAMs, mAb-CD163-PDNPs can actively enrich drugs at the tumor site and have a strong inhibitory effect on TAMs and tumor cells. The result of the in vivo test also shows a good therapeutic effect, with a tumor inhibition rate of 81%. This strategy of delivering anticancer drugs in TAMs provides a new way to develop targeted drugs for immunotherapy of malignant tumors.

## 1. Introduction

With decades of research in the field of cancer, a variety of cancer treatment methods has been developed, among which immunotherapy has opened up new avenues for cancer treatment, and tumor clearance can be achieved by regulating other cells [1,2,3,4]. The core of immunotherapy is to use the immune system to eradicate diseased cells or protect healthy cells, and fight against tumors through countless coordinated pathways, thus triggering a lasting tissue-specific response [5,6,7,8,9,10,11]. Infiltrating immune cells are rich components of solid tumors [12]. The complex balance between pro-tumor and anti-tumor effects caused by the penetration of immune cells forms a chronic inflammatory microenvironment that is critical for tumor growth, progression, and invasion [13,14,15,16]. Macrophages play a vital role in the tumor immune microenvironment [17,18]. Based on morphological, phenotypic, and functional heterogeneity, macrophages can be divided into two different types: classic activated M1-type and alternative activated M2-type macrophages. M1 plays a key role in anti-tumor immunity, mainly mediating the pro-inflammatory processes in the tumor microenvironment, while M2 has been proved to have pro-tumor function and promote tumor growth and metastasis [19,20].

Tumor-associated macrophages (TAMs) are the most diverse immune cells in the tumor microenvironment (TME) and are critical for tumor growth [21]. Due to the role of various chemokines (such as CCL2 and CCL5) and cytokines (for example, CSF1 and members of the VEGF family) at the tumor site, monocytes are recruited at the tumor site to form TAMs [22,23]. The lactic acid produced by the enhanced glycolytic activity of cancer cells further leads to the acidification of TME, induces the regulation of macrophages through G protein-coupled receptors (GPCRs) and IL-1β converting enzyme (ICE), enhances the expression of VEGF and arginase, thus promoting the M2-type features of TAMs to induce the regulation of macrophages [24,25,26]. In recent decades, TAMs have attracted much attention for their powerful ability to inhibit or promote tumor development. A large number of studies have shown that the high density of M2-type macrophages is related to tumor cell proliferation, angiogenesis, immunosuppression, drug resistance, induced histologic malignancy, and poor clinical prognosis [27,28].

In recent years, immunotherapy targeting TAMs has been developed [29,30]. The surface of M2-type macrophages overexpresses some specific receptors, such as CD206 and CD163 [31]. Since M2-type macrophages are the main components of TAMs, researchers have used these receptors as targets to propose a series of tumor immunotherapy strategies for TAMs. At present, the studies in this field mainly focus on blocking recruitment of TAMs, depleting TAMs, reversing the phenotype and enhancing the immune function of TAMs [32,33,34]. Among them, clearing TAMs from tumor sites is a good way to inhibit tumor development, which can be achieved by inducing apoptosis of TAMs. For example, when using phosphonate to deplete TAMs, such drugs can directly target TAMs, causing damage to TAMs, leading to TAMs depletion, and achieving antitumor purposes. Liu et al. [35] developed a biodegradable lipid-coated nanoparticle to improve the efficacy of radiation therapy. The nanoparticles can directly consume TAMs in the body and inhibit angiogenesis, thereby achieving significant anti-tumor effects. Miller et al. [36] designed a therapeutic regimen that used TAMs as a sustained-release library of platinum (IV) precursors for nanotherapy. Nanodrugs accumulate at a high level in TAMs, which act as “cell drug reservoirs”. In fact, TAMs release Pt payloads into adjacent tumor cells over time.

Doxorubicin (DOX) is a topoisomerase II inhibitor, which can partially hinder the efficient repair of damaged DNA. It has been used for more than 40 years and is still one of the most commonly used drugs in much cancer chemotherapy [37]. In tumor cells, DOX induces apoptosis by blocking the process of replication and transcription, generating free radicals to destroy cell membranes, proteins, etc. [38]. If DOX can be delivered to TAMs, which acts as a DOX repository, it is possible to increase the accumulation of TAMs at tumor sites while consuming TAMs. However, cardiotoxicity limits its long-term use [39]. To overcome this limitation, encapsulating it in smart drug delivery systems (DDS) can significantly reduce this unnecessary toxicity compared to free administration (i.e., traditional anti-cancer drug treatment) [40,41]. In our previous studies, we tried to use polyphosphoester-loaded DOX prodrug to prepare nanoparticles, and then linked antibodies on their surface to obtain good cytostatic effects [42]. Nanoparticles with polyethylene glycol (PEG) on the surface can effectively reduce their binding to proteins [43,44,45,46] and have the probability of avoiding recognized and cleared by the endothelial reticular system (RES).

In this work, we prepared a polymer prodrug with acid responsiveness in a tumor microenvironment to deliver DOX. As shown in Figure 1, the PEG block was used as a hydrophilic segment, and the atom transfer radical polymerization (ATRP) of a hydrophobic monomer 2-(4-formylbenzoyloxy)ethyl methacrylate (FBEMA) was conducted to obtain an amphiphilic polymer N_3_-PEG-*b*-(PFBEMA). DOX was linked with the aldehyde group of the PEBEMA side group via Schiff base reaction to produce a polymeric prodrug N_3_-PEG-*b*-(PFBEMA-DOX) with stable drug loading and pH responsiveness. It can self-assemble into nanoparticles in aqueous solution. In another reaction, dibenzocyclooctynyl-coupled CD163 monoclonal antibody (mAb-CD163-DBCO) was prepared by amidation between NHS-PEG_4_-DBCO and mAb-CD163. Finally, mAb-CD163-PDNPs nanoparticles were obtained through the “Click” reaction between the azide group (-N_3_) on the surface of N_3_-PEG-*b*-(PFBEMA-DOX) and the dibenzocyclooctynyl group (-DBCO) of mAb-CD163-DBCO. In this strategy, the nanoparticles are used for TAMs because they can target a large number of TAMs recruited at the tumor site to deliver anti-cancer drug DOX, which can effectively depleting TAMs, and can also be served as a warehouse of nanoparticles to enhance drug retention at the tumor site and kill tumor cells.

## 2. Materials and Methods

### 2.1. Materials and Characerization

The following agents were purchased and used as received: doxorubicin hydrochloride (DOX HCl, 99%, Beijing Zhongshuo Pharmaceutical Technology Development, Beijing, China), α-bromoisobutyryl bromide (98%, Sigma-Aldrich, Shanghai, China), N_3_-PEG-OH (*M*_n_ = 5000 g·mol^−1^, Shanghai Ponsure Biotech, Inc., Shanghai, China), CD163 mouse monoclonal antibody (Invitrogen, Catalog 14-1631-82, Xi’an, China), dibenzocyclooctynyl-tetrapolyethylene glycol-succinimide ester (NHS-PEG_4_-DBCO, Xi’an Ruixi Biological Technology Co., Ltd., Xi’an, China), 4-formylbenzoic acid (98%, Adamas-beta, Shanghai, China), 2-hydroxyethyl methacrylate (96%, Shanghai Aladdin Biochemical Technology Co., Ltd., Shanghai, China), CuBr (98%, J&K Chemical, Shanghai, China), *N*, *N′*, *N′*, *N″*, *N″*-pentamethyl diethylenetriamine (PMDETA, 98%, Sigma-Aldrich, Shanghai, China), methyl thiazolyl tetrazolium (MTT, 98%, Sigma-Aldrich, Shanghai, China), cell counting kit-8 (CCK-8, Beijing Labgic Technology Co., Ltd., Shanghai, China). Triethylamine (TEA, A.R., Enox, Changshu, China), dimethyl sulfoxide (DMSO, A.R., Enox, Changshu, China) and *N,N*-dimethylformamide (DMF, A.R., Enox, Changshu, China) were distilled before use, dichloromethane (CH_2_Cl_2_, A.R., Enox, Changshu, China) was dried over CaH_2_ for at least 24 h and distilled before use. Milli-Q water (18.2 MΩ.cm at 25 °C) was produced through a water purification system (Simplicity UV, Millipore, Shanghai, China). Other reagents for biological experiments were analytical reagents and used as received unless otherwise mentioned. Detailed characterization methods are provided in the Appendix A.

### 2.2. Synthesis of FBEMA Monomer

The hydrophobic monomer 2-(4-formylbenzoyloxy)ethyl methacrylate (FBEMA) was synthesized according to previous literature [47]. Specifically, to a 250 mL branch tube flask, 4-formylbenzoic acid (7.50 g, 0.05 mol), DMAP (1.22 g, 0.01 mol), and EDCI (11.50 g, 0.06 mol) were added under a nitrogen atmosphere and dissolved with 70 mL of CH_2_Cl_2_. We put the flask in an alcohol bath at 0 °C in nitrogen atmosphere. Hydroxyethyl methacrylate (6.5 g, 0.05 mol) was dissolved in 15 mL of CH_2_Cl_2_ and transferred to a constant pressure drip funnel, then slowly dropped into the reaction flask and stirred at the same time. After dropping, we transferred the reaction flask to an oil bath at 30 °C and continued the reaction for 48 h. The crude product was extracted twice with 1 M HCl solution, saturated NaHCO_3_ solution, and saturated NaCl solution, respectively. The organic phase was collected. The crude product was dried with anhydrous Na_2_SO_4_ for 4 h to remove residual water, then concentrated to about 10 mL. Column chromatography was used to purify the crude product with a silica gel 200–300 mesh and an eluent of ethyl acetate: petroleum ether (5:2 by volume). We took the first product point and removed the solvent in vacuum for 24 h to obtain a pale yellow oil (10.38 g, yield 74.1%).

### 2.3. Synthesis of PEG Macromolecular ATRP Initiator

To a 50 mL branch flask, 0.23 g (1.0 mmol) of α-bromoisobutyryl bromide was added and dissolved in 10 mL of anhydrous THF. N_3_-PEG-OH (1.00 g, 0.2 mmol) was dissolved in 20 mL anhydrous THF, and TEA (0.04 g, 0.4 mmol) was mixed to the above solution. Then, we slowly dripped the above solution into the branch flask under a nitrogen atmosphere of 25 °C, stirring for 12 h. After the reaction was completed, the triethylamine salt was removed by filtration, and the product was concentrated to about 3 mL. The product was precipitated three times with 100 mL of n-hexane and dried under vacuum for 24 h to obtain a pure white powder (0.927 g, yield 92.7%).

### 2.4. Synthesis of N_3_-PEG-b-PFBEMA Amphiphilic Polymer

We weighed 0.60 g (2.29 mmol) of FBEMA monomer, 0.50 g (0.1 mmol) of PEG macromolecular ATRP initiator, and 0.10 g (0.58 mmol) of PMDETA, dissolved them in 5 mL of DMF and transferred to a dry Schlenk bottle. Oxygen was removed from the system through a freeze-extraction cycle. After that, nitrogen was introduced to maintain the bottle in a nitrogen atmosphere. CuBr (0.04 g, 0.28 mmol) was quickly weighed and added to the bottle, frozen and pumped again, then transferred to an oil bath at 80 °C for 12 h. After the reaction was completed, the sample solution in the bottle was transferred to a dialysis bag (MWCO 6000), dialyzed with DMF for 24 h, and then dialyzed in deionized water for 24 h. Heavy metal adsorption ion resin was added to remove Cu^2+^ ions, and finally, freeze-dried under vacuum to obtain a pale yellow amphiphilic polymer N_3_-PEG-*b*-PFBEMA (0.657 g, yield 59.7%).

### 2.5. Synthesis of N_3_-PEG-b-(PFBEMA-DOX) Polymer Prodrug

To a 50 mL dry flask, we added 80 mg of N_3_-PEG-*b*-PFBEMA, 20 mg of DOX·HCl and 30 mg of anhydrous TEA and added 5 mL of anhydrous DMSO to dissolve the samples. The reaction was carried out in a nitrogen atmosphere at 35 °C for 12 h. After the reaction, the sample solution was transferred to a dialysis bag (MWCO 7000 Da). The unreacted DOX was first dialyzed in DMSO for 24 h, then dialyzed in phosphate buffer solution with pH 7.4 (PB 7.4) for 24 h to replace DMSO, and finally, freeze-dried under vacuum to obtain a red solid (95 mg, yield 95%).

### 2.6. Synthesis of mAb-CD163-DBCO

The mAb-CD163-DBCO was prepared by amidation reaction of NHS-PEG_4_-DBCO with the amino group on the CD163 monoclonal antibody (mAb-CD163), and the molar ratio of mAb-CD163 to NHS-PEG_4_-DBCO was set to 1:30. Specifically, 100 μL of mAb-CD163/PB 7.4 solution with an mAb concentration of 0.5 mg·mL^−1^ was added to 8 μL of NHS-PEG_4_-DBCO in DMSO solution (1.0 mg·mL^−1^) and incubated for 12 h at 25 °C. After the reaction, the unreacted NHS-PEG_4_-DBCO was removed by centrifugation using an ultrafiltration tube (MWCO 10,000 Da, 7000 g, 5 min), and washed three times with PB 7.4 solution to obtain mAb-CD163-DBCO. The number of DBCO grafting on each monoclonal antibody was measured by matrix-assisted laser desorption/ionization time-of-flight mass spectrometry (MALDI-TOF-MS).

### 2.7. Preparation and Characterization of mAb-CD163-PEG-b-(PFBEMA-DOX) Prodrug Nanoparticles

Firstly, N_3_-PEG-*b*-(PFBEMA-DOX) prodrug nanoparticles were prepared by the solvent replacement method. A measure of 12.5 mg of N_3_-PEG-*b*-(PFBEMA-DOX) was dissolved in 1.0 mL DMF, then the solution was transferred to a 50 mL round-bottom flask. A measure of 18 mL of PB 7.4 buffer solution was added into the flask using a microinjection pump at a rate of 2 mL·h^−1^ and stirred while adding. After addition, the solution was stirred for 5 h, then transferred to the dialysis bag (MWCO 7000 Da) and dialyzed in PB 7.4 solution for 4 h to replace DMF. The solution was diluted to 25 mL to obtain N_3_-PEG-*b*-(PFBEMA-DOX) prodrug nanoparticles (abbreviated as PDNPs) solution with the prodrug concentration of 0.5 mg·mL^−1^. After that, mAb-CD163-DBCO was added and reacted in a shaker at 25 °C and 120 rpm for 12 h to yield mAb-CD163-PDNPs. Unbonded mAb-CD163-DBCO was removed by ultracentrifugation (210,000 g, 4 °C, 50 min). After freeze drying, the supernatant was retained and dissolved in PB 7.4 solution. The content of mAb-CD163-DBCO in the supernatant was determined by BCA protein concentration assay. Then, the centrifuged nanoparticles were allowed to stand and diluted to an appropriate concentration with a buffer solution of pH 7.4. Dynamic light scattering (DLS) and transmission electron microscopy (TEM) were used to characterize the particle size and morphology of the nanoparticles.

### 2.8. Self-Assembly Behavior

The critical aggregation concentration (CAC) of PEG-*b*-(PFBEMA-DOX) prodrug was determined by the pyrene fluorescent probe method. Briefly, we added 50 μL of pyrene acetone solution in groups and removed acetone under vacuum. Then, we added 5mL of polymer aqueous solution of different concentrations (the concentration of pyrene in this solution was 6 × 10^−7^ M), sonicated at room temperature for 30 min, and stirred for 48 h. The fluorescence intensity of pyrene was analyzed by a fluorescence spectrophotometer. The excitation wavelength was set to 335 nm, which was determined at a slit width of 5 nm in the emission spectrum in the wavelength range of 350 nm to 550 nm. The intensity ratio (*I*_3_/*I*_1_) of the peak (383 nm, *I*_3_) to the peak (372 nm, *I*_1_) in the emission spectrum was analyzed as a function of the concentration of polymer precursors. The intersection point was determined as the CAC value.

### 2.9. Stability and In Vitro Drug Release

To study the stability of the mAb-CD163-PNDPs nanoparticles, the sample solutions with different concentrations were configured, stored at 4 °C, and 10% fetal bovine serum was added. The particle size changes were measured and recorded daily. These samples were also dissolved in buffer solutions of different pH media and the changes of particle size measured to study their pH stability. To determine the pH-sensitive DOX release profile of the samples, we added mAb-CD163-PDNPs to buffer solutions of different pH and shook horizontally at 120 rpm at 37 °C. We collected 5 mL of buffer outside the dialysis bag at predetermined intervals and replenished an equal amount of buffer. A fluorescence spectrophotometer was used to determine the release of DOX at different pH media within 76 h.

### 2.10. In Vitro Hemolysis Activity

The blood vessels were pretreated with heparin sodium. The mouse eyeballs were removed, and the blood flowed down the centrifuge tube wall, shaking evenly to prevent clotting. We added 2 mL of PBS, gently blew well with a pipette gun, and centrifuged to pellet the red blood cells (500 g, 10 min). After aspirating the supernatant, the red blood cell suspension was transferred to a 15 mL centrifuge tube and 10 mL of PBS was added. After that, the supernatant was centrifuged and removed. This was repeated three times until the supernatant became colorless. We added PBS to make the red blood cell suspension to 10 mL and left it for standby. We added 0.5 mL of red blood cell suspension to each of the 12 clean centrifuge tubes. A measure of 0.5 mL of mAb-CD163-PDNPs of different concentrations was added into five of the tubes. To the other five tubes, 0.5 mL of DOX with different concentrations was added, where the concentration of DOX corresponds to the group of mAb-CD163-PDNPs. In addition, the last two tubes were added with PBS and Milli-Q water, respectively, for the negative control group and the positive control group (the samples were dissolved with PBS). We shook the centrifuge tube (37 °C, 200 rpm) for 3 h. We pelleted intact red blood cells through centrifugation (7000 g, 4 °C, 5 min), removed 3 × 100 μL of supernatant from each centrifuge tube and added them to 96-well plates with absorbance of hemoglobin in the supernatant at 540 nm by a microplate reader. The hemolysis percentage was calculated according to Equation (1):(1)$$\mathrm{Hemolysis}\;(\%)=\frac{\mathrm{ODsample}-\mathrm{ODnegative}\;\mathrm{control}}{\mathrm{ODpositive}\;\mathrm{control}-\mathrm{ODnegative}\;\mathrm{control}}\times 100$$
where OD_sample_, OD_negative control_, and OD_positive control_ represent the OD values of the wells treated with samples. The negative control wells with PBS and the positive control wells with Milli-Q water, respectively.

### 2.11. Cell Culture

Human umbilical vein endothelial cells (HUVEC cells), mouse hepatoma cells (H22 cells), and mononuclear macrophageleukemia (RAW264.7 cells) were obtained from American Type Culture Collection (ATCC) and cultured in high glucose DMEM containing 10% fetal bovine serum (FBS) and 1% penicillin/streptomycin solution. Both cell lines were passaged once every 2 days and incubated at 37 °C in an atmosphere containing 5% CO_2_ and certain humidity. Among them, M2-type macrophages were induced by RAW264.7 cells. This was done by adding a certain amount of interleukin 4 (IL-4) to the medium of RAW264.7 cells at a concentration of 20 ng·mL^−1^ for 12 h. Interleukin 10 (IL-10) and Arginase 1 (Arg1) can be used as markers as M2-type macrophages [48], therefore, RAW264.7 cell induction was determined by the mouse IL-10 Elisa kit and Arg1 Elisa kit. Refer to the method in the instructions and quantify the expression of IL-10 and Arg1 in the cell culture supernatant using a microplate reader. To compare the expression of CD163 on the surface of macrophages before and after induction, we performed the assay by the following method: Unpolarized (RAW264.7) and polarized (M2-type) macrophages were gently pipetted off the dish wall with a pipette, respectively. The cell membrane was broken by repeated freeze-thawing three times with liquid nitrogen along with the medium. The supernatant was then collected by centrifugation, where CD163 on the cell membrane was soluble in the supernatant, and the CD163 content in the supernatant was determined using the mouse CD163 Elisa kit.

### 2.12. In Vitro Cytotoxicity

HUVEC cells, H22 cells or M2-type macrophages were seeded into 96-well plates, respectively, at a density of 5 × 10^3^ per well to allow attachment to the plate bottom. Cells were then incubated with corresponding samples at series of concentrations (DOX concentration is set to 0.01 to 5 mg·L^−1^ for a total of 10 groups, *n* = 3). The cytotoxicity was measured using the MTT or CCK-8 assay.

### 2.13. Cellular Uptake

The cellular uptake and intracellular release in M2-type macrophages of free DOX, PDNPs, and mAb-CD163-PDNPs were investigated by the confocal laser scanning microscope (Zeiss, Jena, Germany, LSM 800). Briefly, cells were loaded in an *Φ* 20 mm cell culture dish at a density of 1 × 10^5^ cells per dish and incubated for 12 h. Then, the culture medium was removed and replaced by the fresh medium with H 33342 (H 33342/DMEM = 1/1000, *v*/*v*), cultured for 30 min, then washing with PBS three times. Finally, the medium involving 10% free DOX, PDNPs, and mAb-CD163-PDNPs were added with the same DOX content (DOX concentration was 6 mg·L^−1^), respectively. Cell culture dishes were moved into the culture system of a live cell imaging system, which maintained an environment of 37 °C with 5% CO_2_. Then, we obtained images in 5 h at excitation wavelengths of 480 nm (red) and 340 nm (blue).

### 2.14. Animal Models

Six to eight weeks old balb/c mice were purchased from Hangzhou Ziyuan Laboratory Animal Technology Co., Ltd. (Hangzhou, China). Animals were housed according to AAALAC (Association for Assessment and Accreditation of Laboratory Animal Care) guidelines. Subcutaneous H22 tumor model was established by injecting 1 × 10^6^ H22 cells under the skin of the right back leg. Orthotopic H22 tumor model was established by surgically transplanting H22 tumor mass on the back of balb/c mice.

### 2.15. In Vivo Antitumor Efficacy

To assess the in vivo antitumor efficacy of mAb-CD163-PDNPs, 24 balb/c mice carrying H22 tumors were randomized into four groups (*n* = 6). When the average tumor volume reaches about 200 mm^3^, these mice were treated with different types of therapy. In the first control group, mice received intraperitoneal injection of PBS every two days. In the remaining three groups, free DOX, PDNPs, and mAb-CD163-PDNPs were injected into mice at doses of 5 mg·kg^−1^ DOX equivalents every two days, respectively. All mice were injected seven times in a row over a two-week period. Tumor diameter was recorded and mouse body weighted before each injection. We calculated the tumor volume according to the formula: V = 0.5 × L × W^2^ (where L is the longest diameter and W is the shortest diameter). After 14 days, mice were sacrificed and recorded. Tumors and major organs, including heart, liver, spleen, lungs, and kidneys were collected and cleaned with PBS for biodistribution analysis.

### 2.16. In Vivo Biodistribution and Fluorescence Imaging

In vivo biodistribution experiments were performed after completion of treatment. After all mice (balb/c) were sacrificed, their tumors and major organs (heart, liver, spleen, lungs, and kidneys) were removed and fluorescence images of these organs and tumors were tested.

### 2.17. Histopathology and Immunohistochemical Analysis

For histological evaluation, after the anti-tumor experiment, we dissected groups of mice to obtain tumor tissue, which were then treated with paraffin embedding. After dewaxing with xylene, we cut the tumor tissue into 5 μm sections. Tissue sections were stained with H&E. Finally, histological changes and apoptosis were observed by light microscopy.

Immunohistochemical analysis was based on the operation guidelines of the immunohistochemistry kit, and immunohistochemical staining of TNF-α, IFN-γ, IL-10, and IL-12 of each group of tumor sections. Finally, the expression of each cytokine was observed and photographed with a normal light microscope.

### 2.18. Statistical Analysis

All experiments were conducted with at least three or more parallel samples, and the statistical data were evaluated via GraphPad Prism 7.0 software. The data were presented in terms of mean ± standard deviation (mean ± SD). In addition, single factor variance analysis (ANOVA) was used to test the significant difference between the data, which were reflected by using * *p* < 0.05, ** *p* < 0.01, *** *p* < 0.001, and **** *p* < 0.0001.

## 3. Results and Discussion

### 3.1. Synthesis and Characteristics of N_3_-PEG-b-(PFBEMA-DOX)

The pH-responsive deblock copolymer N_3_-PEG-*b*-PFBEMA was prepared via ATRP reaction of FBEMA monomer using N_3_-PEG-Br as the macromolecular initiator. Then, the Schiff base reaction occurred between the aldehyde group of the PEBEMA side group and the -NH_2_ of DOX, which makes DOX bond with the amphiphilic block copolymer N_3_-PEG-*b*-PEBMEA through the aldoxime bond chain to obtain the amphiphilic polymer prodrug N_3_-PEG-*b*-(PFBEMA-DOX). Figure 2 shows the synthesis procedure.

As shown in Figure 1, the successful synthesis of diblock copolymer N_3_-PEG-*b*-PFBEMA and the prodrug N_3_-PEG-*b*-(PFBEMA-DOX) were verified by ^1^H NMR. The ^1^H NMR spectra of the monomer (FBEMA) and the macromolecular initiator (N_3_-PEG-OH) are shown in Appendix A. In addition, the molecular weights of the polymers were further characterized. In Figure 2a, through the matrix assisted laser desorption ionization time of flight mass spectrometry (MALDI-TOF MS), we can confirm the modification of the ATRP macromolecular initiator. Then, the polymer chains with different hydrophobic segment lengths of PFBEMA were obtained by controlling the initial feed ratios, and the molecular weights of the copolymers were proved by gel permeation chromatography (GPC) (Appendix A). It can be seen that the GPC curve of the copolymer sample after polymerization moves forward significantly, indicating that the molecular weight increases and polymerization occurs successfully. From the change of molecular weight in Appendix A, it can be seen that the molecular weight of the copolymer increased with the increase of the feed ratio, and all the molecular weight distributions are narrow.

After the drug was connected to the main chain of the polymer with different molecular weights, the successful loading of the drug was proven by UV-vis spectroscopy. In Figure 2b, the UV-vis spectrum shows that the diblock copolymer N_3_-PEG-*b*-PFBEMA had no characteristic absorption, while the polymer prodrug N_3_-PEG-*b*-(PFBEMA-DOX) has a significant red shift compared with free DOX. This is due to the change in the chemical environment of DOX and the polymer backbone by imine bonding. In the FT-IR spectra (Figure 2c), we can also observe that the polymer prodrug N_3_-PEG-*b*-(PFBEMA-DOX) shows a distinct -OH peak at 3300 cm^−1^, which is attributed to DOX, indicating that the drug has been successfully loaded. Then, the drug loading was measured by fluorescence spectrophotometer. Table 1 shows the specific drug loading data. The results showed that the drug loading increased with the increase of hydrophobic segment, which was in line with expectations. In these four groups of samples, according to the drug loading data, we selected the fourth sample N_3_-PEG-*b*-(PFBEMA-DOX)-4 for the follow-up experiment.

### 3.2. Modification of CD163 Monoclonal Antibodies

As shown in Figure 3, we modified the CD163 monoclonal antibody to obtain mAb-CD163-DBCO. According to the MALDI-TOF MS in Figure 2d, the calculated result is that each CD163 monoclonal antibody was bound to 3.1 DBCO molecules on average. mAb-CD163-PDNPs were then prepared by “Click” reaction between the azide group (-N_3_) and dibenzocyclooctynyl group (-DBCO). The advantages of this method are high efficiency and a metal-free catalyst, which reduces toxicity. After that, the modification of monoclonal antibodies on the nanoparticles surface was measured, and different feeding ratios were set. As shown in Table 2, the modification efficiency of each group for samples was around 90%, and the feeding ratio of the second group (mAb-CD163-PDNPs-2) was chosen for the subsequent synthesis and experiments considering the cost and practical situation.

### 3.3. Stability of Nanoparticles

After the sample was prepared, we were first concerned about its stability. Critical aggregation concentration (CAC) is an important indicator to evaluate nanoparticle formation and is the lowest concentration required to reveal the formation of nanoparticles by amphiphilic polymers. In this work, we used the pyrene fluorescent probe method to determine the CAC of N_3_-PEG-*b*-(PFBEMA-DOX). Due to the low solubility of pyrene in aqueous solution (~7 × 10^−7^ mol L^−1^), it is preferentially enriched in the hydrophobic region during the assembly of the amphiphilic polymer, so that the fluorescence emission spectrum of pyrene will change significantly, which can be expressed by the ratio of the third emission peak and the first emission peak of pyrene (*I*_3_/*I*_1_). Changes in *I*_3_/*I*_1_ values in pyrene emission spectra can therefore be used to track nanoparticle formation and determine CAC values. Appendix A shows the relationship between *I*_3_/*I*_1_ and the concentration of N_3_-PEG-*b*-(PFBEMA-DOX). The CAC value obtained by linear fitting is 0.032 mg mL^−1^, indicating that samples can aggregate at lower concentrations to form nanoaggregates.

Figure 3 shows the size distribution of mAb-CD163-PNDPs nanoparticles in different environments and the corresponding TEM images. As shown as Figure 3a,b, under the condition of phosphate buffer solution with pH 7.4 (PB 7.4), the size of nanoparticles was about 105 nm, the distribution of nanoparticles was narrow, and the morphology was regular. Figure 3c shows that the particles size did not change significantly within 48 h, and Figure 3d is the state of nanoparticles in PB 7.4. We did not observe nanoparticles aggregation during laser irradiation, indicating that the nanoparticles were relatively stable in solution. In contrast, the nanoparticles are obviously unstable and dissociated in PB 5.0 (Figure 3e,f), which is just conducive to drug release under acidic conditions.

The concentration and serum stability of the mAb-CD163-PDNPs nanoparticles were also investigated. Appendix A shows the size distribution of nanoparticles at different concentrations. From this set of data, it can be seen that even at very low concentrations, the nanoparticles still maintained a relatively stable particle size and particle size distribution. Appendix A shows the size distribution of mAb-CD163-PDNPs after being stirred in PB 7.4 buffer solution containing 10% fetal bovine serum for 48 h, and overall, it is relatively stable.

### 3.4. In Vitro Drug Release

The in vitro drug release behavior of mAb-CD163-PDNPs nanoparticles was studied. As shown in Figure 4a, the drug release was carried out under the conditions of PB 5.0, PB 6.0, and PB 7.4, respectively. It was found that the drug release amount was low under the normal physiological conditions of PB 7.4, while DOX could be rapidly released under the weak acidic conditions of PB 5.0 and PB 6.0. This is because the polymer prodrug N_3_-PEG-*b*-(PFBEMA-DOX) is connected by the imine bond formed by Schiff-base reaction between the aldehyde group of PFBEMA and the amino group of DOX, which can be hydrolyzed under weak acidic conditions to make the drug break away from the polymer chain [49,50]. Drug shedding may destabilize the prodrug nanoparticles, which also explains the dissociation of the nanoparticles under PB 5.0 conditions.

### 3.5. Hemocompatibility

The drug-loaded nanoparticles usually need to undergo a period of blood circulation before reaching the tumor site, which requires that the nanoparticles will not cause damage to red blood cells. If drug-loaded nanoparticles combine with red blood cells, it may lead to higher hemolysis rate (>5%) [51]. The percentage of hemolysis indicates the degree of damage of the erythrocyte membrane by the material. We compared the hemolysis rate of and blood compatibility of free DOX and mAb-CD163-PDNPs on red blood cells.

Figure 4b is a photo of the samples before the hemocompatibility test. It can also be clearly seen that there is no obvious hemolysis in the experimental group compared with the control group, and most of the red blood cells are concentrated at the bottom of the centrifuge tube. In fact, the red in the supernatant was the color of DOX.

As can be seen from Figure 4c, with the increase of DOX concentration, the hemolysis rate of the free DOX group reached about 7%, while the mAb-CD163-PDNPs prodrug group remained below 5%, indicating that the combination of drugs and polymers can reduce the toxic and side effects of drugs to a certain extent. Of course, this is also due to the good biocompatibility of PEG in the copolymer, which makes it difficult for nanoparticles to destroy erythrocytes.

### 3.6. In Vitro Cytotoxicity

The cytocompatibility and cytotoxicity of the samples were then investigated using MTT or CCK-8. First, the supernatant in the macrophage dish was collected before assessing cytotoxicity, then the induction effect of macrophages was determined by ELISA kit. It can be seen in Appendix A that after IL-4 induction, the expression of IL-10, Arg1, and CD163 in M2-type macrophages increased significantly, indicating that most macrophages were induced to M2-type. The cytocompatibility of the copolymer N_3_-PEG-*b*-PFBEMA was then evaluated. As shown in Figure 4d, after co-culture of the N_3_-PEG-*b*-PFBEMA with HUVEC cells, H22 cells, and M2-type macrophages for 48 h, respectively, the survival rate of cells remained almost at 90% or above, which proves that the drug carrier N_3_-PEG-*b*-PFBEMA we synthesized has good biocompatibility.

The cytotoxicity of nanoparticles on H22 cells and M2-type macrophages was verified by cytotoxicity experiments. We investigated the inhibition of free DOX, PDNPs, and mAb-CD163-PDNPs, respectively, against both H22 cells by CCK-8 and M2-type macrophages by MTT assay. Figure 5 shows the relationship between cell viability and sample concentration. We found that cell viability was dependent on sample concentration. Moreover, to further evaluate the inhibitory effect of nanoparticles on cells, the half-maximal inhibitory concentration (IC_50_) values are listed in Table 3. We determined the IC_50_ values of H22 cells (0.166 mg·L^−1^) and M2-type macrophages (0.235 mg·L^−1^), respectively. Compared with free DOX, the IC_50_ values of PDNPs for H22 and M2-type macrophages were 0.367 mg·L^−1^ and 0.860 mg·L^−1^, respectively, which increased overall. Combined with previous hemocompatibility data as well as cytocompatibility data of the diblock copolymer N_3_-PEG-*b*-PFBEMA, we speculated that this may be due to the encapsulation of the biocompatible polymer that reduced the biotoxicity of the drug. At the same time, we found that H22 cells have a slightly lower IC_50_ than M2-type macrophages, which may be due to the lower tolerance to DOX of H22 cells.

Compared with the IC_50_ value of mAb-CD163-PDNPs (0.541 mg·L^−1^) and PDNPs (0.860 mg·L^−1^) on M2-type macrophages, it was not difficult to find that mAb-CD163-PDNPs have a stronger inhibitory effect on M2-type macrophages, which is due to the overexpression of CD163 receptors on the surface of M2-type macrophages, and through the specific recognition between receptors and antibodies, which further confirms that mAb-CD163-PDNPs can be highly enriched in M2-type macrophages and has a stronger inhibitory effect.

### 3.7. Cell Uptake

The enrichment process of the drug in tumor cells was observed by cell uptake assay. Figure 6 shows endocytosis of M2-type macrophages incubated with mAb-CD163-PDNPs, PDNPs, and free DOX, respectively, where blue is the nucleus stained with H 33342 fluorescent dye, and red is the fluorescence of the DOX drug. When mAb-CD163-PDNPs and PDNPs were added to confocal dishes for 1 h, there was no significant fluorescence of drug in the nucleus, indicating that the nanoparticles were not enriched in large quantities in the cells at this time. When the incubation time was extended to 5 h, the red fluorescence in M2-type macrophages incubated with mAb-CD163-PDNPs was significantly enhanced, while the red fluorescence in the PDNPs group increased slightly. Red fluorescence was the weakest among M2-type macrophages incubated with free DOX. These phenomena indicate that after modification with CD163 monoclonal antibody, the polymeric prodrug nanoparticles are more easily endocytosed by M2-type macrophages, which may increase the drug concentration at the tumor site and improve the anti-tumor effect.

### 3.8. In Vivo Antitumor Effect

We had verified that mAb-CD163-PDNPs can actively target M2-type macrophages, enhancing drug enrichment at the tumor site. Therefore, the in vivo therapeutic effect of this nanomedicine was further studied. The nanoparticles were injected into mice through the peritoneal cavity and administered every two days. As shown in Figure 7a, after 7 consecutive doses, the free DOX group (723 mm^3^) showed some tumor suppression compared with the tumor volume (1882 mm^3^) in the PBS group, while the PDNPs group (561 mm^3^) showed a stronger inhibitory effect with a tumor suppression rate of 70%. This may be because the nanoparticles are more likely to be enriched at the tumor site, increasing drug utilization.

In contrast, the mAb-CD163-PDNPs group (362 mm^3^) showed the strongest tumor suppressor effect, reaching a tumor suppression rate of 81%. Representative images of the tumor were taken at the end of treatment (Figure 7b) and the tumor was weighed (Figure 7c). As mentioned above, these results can be attributed to the ability of nanomedicines to enhance cellular uptake. After modifying CD163 monoclonal antibody, the polymer prodrug mAb-CD163-PDNPs cleared a large number of M2-type macrophages in the tumor site, which relieved the immunosuppressive effect of the tumor site to a certain extent, and further enhanced the enrichment of the drug in the tumor site and improved the efficacy.

In addition, body weight changes of mice were recorded (Figure 7d). We found that the average body weight of mice in all groups except the free DOX group did not show a significant downward trend, indicating that both the mAb-CD163-PDNPs and PDNPs prodrug delivery platform showed low systemic toxicity. In the middle and late stages of treatment in the free DOX group, the body weight of mice began to decline, and it was observed that the mice in this group had messy hair and were not in a very good state. After drug treatment, mice in each group were dissected, tumor and various organs (heart, liver, spleen, lung, kidney) were removed, and fluorescence imaging analysis was performed (Figure 7e). The results were consistent with expectations. As previously expected, compared with the free DOX and PDNPs groups, the mAb-CD163-PDNPs group had the highest concentration at the tumor site. This also provides evidence that mAb-CD163-PDNPs group showed better efficacy. In addition, some drugs were also enriched in the liver and kidney of each group. The drug concentration in the liver of the free DOX is higher than that of the other two groups, which may be because the free DOX is easier to accumulate in the liver.

### 3.9. Histopathology and Immunohistochemical Analysis

A series of histological staining assays were further used to assess the antitumor activity of mAb-CD163-PDNPs. As shown in Figure 8, when the tumor slices were observed under microscope, we observed a large number of tumor cells (marked by white arrows) and cell debris (marked by orange arrows). Compared with the normal tumor growth state of the PBS group, the experimental group shows different degrees of tissue necrosis, nuclear contraction, nuclear fragmentation, or other pathological features. In particular, the mAb-CD163-PDNPs group exhibits the highest degree of tissue necrosis, as evidenced by the large number of cell debris in Figure 8d, reflecting the strongest antitumor potency.

To verify whether mAb-CD163-PDNPs are able to improve the immunosuppressive microenvironment at the tumor site, we performed immunohistochemical analysis of the tumor (Figure 9). Among them, TNF-α is currently one of the most effective bioactive factors found to directly kill tumors. IFN-γ has antiviral, immunomodulatory, and antitumor abilities. IL-12 is an antitumor factor secreted by M1-type macrophages, while IL-10 is an immunosuppressive factor produced mainly by M2-type macrophages [52]. Due to the autoimmune effect of the tumor site, the PBS control group had abundant TNF-α secretion (blue part in the figure). Under the action of anti-tumor drugs, causing tissue necrosis, TNF-α has less distribution in necrotic sites for both the free DOX group and PDNPs group, but it is still abundant in normal tissue parts.

Interestingly, the mAb-CD63-PDNPs group shows the richest TNF-α secretion and was relatively evenly distributed. This may be because mAb-CD63-PDNPs specifically deplete M2-type macrophages, to a certain extent, lifting the immunosuppressive effect of the tumor site and causing a stronger immune response. For IFN-γ, results similar to TNF-α have also emerged. Among them, the IFN-γ staining results of the free DOX group and PDNPs group were not obvious, and it was speculated that it may be due to the negative effects of tissue damage. Similarly, for the mAb-CD63-PDNPs group, the IFN-γ staining results of the tissue necrotic part were not obvious, but a richer IFN-γ distribution could be observed in the normal tissue part. For IL-12, it can be seen that the distribution for the PBS control group is very small, while the results of the mAb-CD63-PDNPs group are relatively obvious. IL-10 shows the exact opposite of IL-12. This is because macrophages at the tumor site are mainly M2-type, while IL-12 is mainly produced by M1-type macrophages, and IL-10 is mainly produced by M2-type. Since mAb-CD163-PDNPs deplete a large number of M2-type macrophages, most of the newly recruited macrophages at the tumor site are M1-type, which led to the above results, indicating that the mAb-CD63-PDNPs group has a better immune enhancement.

## 4. Conclusions

In summary, we have designed and prepared the CD163 monoclonal antibody modified doxorubicin-polymer prodrug nanoparticles, mAb-CD163-PDNPs, for actively targeting tumor-associated macrophages (TAMs). This strategy enables targeted drug delivery to TAMs and responds to the lower pH of the tumor site to the release drug. While depleting TAMs, TAMs are also used as temporary “drug warehouses” to achieve the effect of drug enrichment at the tumor site. The stability and in vitro drug release experiments show that these nanoparticles are stable under physiological conditions but can effectively dissociate and release DOX under acidic conditions. In the PB 5.0 environment, the final in vitro drug release after 76 h can reach about 80%. At the cellular level, the nanoparticles can be effectively taken up by M2-type macrophages and have a good inhibitory effect on tumor cells and M2-type macrophages. In vivo studies also demonstrate that the mAb-CD163-PDNPs are highly enriched at the tumor site. The final results also show that the mAb-CD163-PDNPs have achieved the highest tumor suppression rate (81%) and the best therapeutic effect, and it also plays a positive role in regulating the immune microenvironment. Overall, this study provides a practical method for targeted drug delivery and tumor immunotherapy: that is, through antibody targeting and polymer prodrug nanoparticles, the delivery of anticancer drugs into TAMs can enhance the enrichment of drugs in the tumor site, activate the immune response to a certain extent, and improve the anti-tumor effect.

## Data Availability

All the data generated within this study were provided in Appendix A.

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
