# Peer review of "CD163 Monoclonal Antibody Modified Polymer Prodrug Nanoparticles for Targeting Tumor-Associated Macrophages (TAMs) to Enhance Anti-Tumor Effects"

_pharmaceutics, 2023, doi:10.3390/pharmaceutics15041241_

Round 1

Reviewer 1 Report

In this article the authors designed TAM-based immunotherapy which is promising strategy. As we know, TAMs, mostly M2-type macrophages, promote tumor development, invasion, and metastasis. Active targeting to TAMs with  CD163-expressing M2-type macrophages is interesting. In this study, the authors prepared CD163 monoclonal antibody modified doxorubicin-polymer prodrug nanoparticles (mAb-CD163-PDNPs) with pH responsiveness and targeted delivery by bonding DOX to the polymer. the platform shows that  TAMs depletion  and enriching medicines at the tumor site, inhibiting TAMs, and tumor cells. In vivo tests demonstrated signficant tumor inhibition and good therapeutic efficacy. 

The authors investigations and experimental works was extinsive and I sugget the manuscript to be accepted; however, there are multiple isuues with articles that can be addressed:

1- Abstract can be improved.

2-In line 256 "Interleukin 10 (IL-10) can be used as a maker as M2-type mac- rophages, therefore, RAW264.7 cell induction was determined by the mouse IL-10 Elisa kit."  Did you mean Marker?

3- In figure 3 and 4  and results 3.3. and  3.4.  What is PB abbreviation is stand for? 

4- Figure 4e. Please explain the results clearly in the text.

5- Results and discussion section should be separeted and adding stand alone discussion would be appropriate with adding more refernces in the discussion section.

6- Some of the figure quality could be improved.

7- I found minor grammatical and typo errors which can be modified. 

Overall, the manuscript is well written and I suggest acceptance of the manuscript after addressing all the comments.

Author Response

Response to Reviewers’ Comments on Pharmaceutics-2270735

Dear Reviewer,

On behalf of our authors, I would like to thank you for your careful review. The following shows your comments and our responses.

Reviewer #1:

General Comments: In this article the authors designed TAM-based immunotherapy which is promising strategy. As we know, TAMs, mostly M2-type macrophages, promote tumor development, invasion, and metastasis. Active targeting to TAMs with CD163-expressing M2-type macrophages is interesting. In this study, the authors prepared CD163 monoclonal antibody modified doxorubicin-polymer prodrug nanoparticles (mAb-CD163-PDNPs) with pH responsiveness and targeted delivery by bonding DOX to the polymer. the platform shows that TAMs depletion and enriching medicines at the tumor site, inhibiting TAMs, and tumor cells. In vivo tests demonstrated significant tumor inhibition and good therapeutic efficacy.

The authors investigations and experimental works was extensive and I suggest the manuscript to be accepted; however, there are multiple issues with articles that can be addressed:

Response: Thank you for your highly encouraging comments. We have revised the indicated issues accordingly, which have been highlighted in yellow in the revised manuscript and also attached below for your consideration.

Comment 1: Abstract can be improved.

Response: Thank you for your comment. We have modified Abstract and added the following sentence: “First, DOX was bonded with the aldehyde group of a polymer by Schiff base reaction to form amphiphilic polymer prodrugs, which could self-assemble into nanoparticles in the aqueous solution. Then, mAb-CD163-PDNPs were generated through a “Click” reaction between the azide group on the surface of the prodrug nanoparticles and dibenzocyclocytyl-coupled CD163 monoclonal antibody (mAb-CD163-DBCO).”

Comment 2: In line 256 "Interleukin 10 (IL-10) can be used as a maker as M2-type macrophages, therefore, RAW264.7 cell induction was determined by the mouse IL-10 Elisa kit." Did you mean Marker?

Response: Thank you for your comment. “Maker” has been changed to “Markers”.

Comment 3: In figure 3 and 4 and results 3.3. and 3.4. What is PB abbreviation is stand for?

Response: PB is an abbreviation for phosphate buffer. We have added the full name of PB where appropriate.

Comment 4: Figure 4e. Please explain the results clearly in the text.

Response: Thank you for your comment. Figure 4e has been changed to Figure 4d. We have explained the results of Figure 4d on Page 14. The following sentence have been added to the revised manuscript: "after co-culture of the N3-PEG-b-PFBEMA with HUVEC cells, H22 cells and M2-type macrophages for 48 h, respectively, the survival rate of cells remained almost at 90% or above, which proves that the drug carrier N3-PEG-b-PFBEMA we synthesized has good biocompatibility."

Comment 5: Results and discussion section should be separated and adding stand alone discussion would be appropriate with adding more references in the discussion section.

Response: Thank you for your suggestion. We have read some articles recently published in Pharmaceutics. Referring to the format of most articles, we put the results and discussion together in order to facilitate the simultaneous presentation of the results and corresponding discussion, so that readers can easily read them. In addition, we have added the necessary references, as Ref. 30, and 48.

Comment 6: Some of the figure quality could be improved.

Response: Thanks for the suggestion. We have adjusted the typography, font size, etc. including Figures 2, 3, 4, 5, 6, 7 and Scheme 2.

Comment 7: I found minor grammatical and typo errors which can be modified.

Response: Thanks for reviewing our manuscript. We have carefully corrected the grammatical and typo errors.

General Comments: Overall, the manuscript is well written and I suggest acceptance of the manuscript after addressing all the comments.

Response: Thank you for your kind comments once again. Your suggestions are very helpful in improving the quality of our manuscripts.

Best regards,

Peihong Ni (Ph.D.)
Professor in Macromolecular Science
College of Chemistry, Chemical Engineering and Materials Science
Soochow University 
Suzhou 215123
China

Reviewer 2 Report

The manuscript of CD163 Monoclonal Antibody Modified Polymer Prodrug Nanoparticles for Targeting Tumor-Associated Macrophages (TAMs) to Enhance Anti-Tumor Effect by Yang et al. is a complete and relevant study that contributes to the advance of immunotherapy. The text is clear, and the results are technically appropriate. I consider that two improvements are necessary.

1. In Figure 6, details of cell images are hard to see and it should be adequate to provide zoom of some cells in each panel.

2. Although the procedures and techniques are described in detail and the results are excellent, it is important to highlight in the abstract and conclusion what is the more significant innovation in this study.

Author Response

Response to Reviewers’ Comments on Pharmaceutics-2270735

Dear Reviewer,
On behalf of our authors, I would like to thank you for your careful review. The following shows your comments and our responses.

Reviewer #2:

General Comments: The manuscript of CD163 Monoclonal Antibody Modified Polymer Prodrug Nanoparticles for Targeting Tumor-Associated Macrophages (TAMs) to Enhance Anti-Tumor Effect by Yang et al. is a complete and relevant study that contributes to the advance of immunotherapy. The text is clear, and the results are technically appropriate. I consider that two improvements are necessary.

Response: We are appreciated for your encouraging comments on our manuscript. We have revised the article as you suggested.

Comment 1: In Figure 6, details of cell images are hard to see and it should be adequate to provide zoom of some cells in each panel.

Response 1: Thank you for your comment. We used a confocal laser scanning microscope to observe the fluorescence intensity of DOX as it enters the cells. Because H 33342 can only stain the nucleus, it seems that the details of the cell image cannot be displayed in the image.

Comment 2: Although the procedures and techniques are described in detail and the results are excellent, it is important to highlight in the abstract and conclusion what is the more significant innovation in this study.

Response 2: Thank you for your kind comments. We have revised the abstract and conclusions. The following sentence has been added to Abstract. “First, DOX was bonded with the aldehyde group of a polymer by Schiff base reaction to form amphiphilic polymer prodrugs, which could self-assemble into nanoparticles in the aqueous solution. Then, mAb-CD163-PDNPs were generated through a “Click” reaction between the azide group on the surface of the prodrug nanoparticles and dibenzocyclocytyl-coupled CD163 monoclonal antibody (mAb-CD163-DBCO).” Another sentence has been added to Conclusion. “This strategy enables targeted drug delivery to TAMs and responds to the lower pH of the tumor site to release drug”, “And it also plays a positive role in regulating the immune microenvironment.”

Best regards,

Peihong Ni (Ph.D.)
Professor in Macromolecular Science
College of Chemistry, Chemical Engineering and Materials Science
Soochow University 
Suzhou 215123
China

Reviewer 3 Report

In this study Yang et al showed the development of a promising anti-TAM therapeutic from initial nanoparticle development through to in vivo analysis. This is an interesting study that utilises a monoclonal antibody as a targeting factor of their nanoparticle with doxorubicin as the cytotoxic payload. While on the whole this looks like a well performed study I have some issues with regards to the evidence of targeting and the need for additional controls.

The paper utilises an antibody that targets CD163, but there is no confirmation that the cell lines tested have (or lack) CD163. For example, in line 483-4 “which is due to the overexpression of CD163 receptors on the surface of M2-type macrophages” there is no evidence CD163 is overexpressed in the induced RAW264.7 cell line. A Western blot, flow cytometry or immunofluorescence experiment is needed to show relative CD163 levels in RAW264.1 cells before and after induction and the control H22 cells. This is especially important as, as far as I can tell, Proteintech group supply two mAb against CD163 and both are validated against the human version of the protein and both RAW and H22 cells are mouse. (Please also state the product number of the antibody used for the paper).

Line 500-4 “These phenomena indicate that after the modification of CD163 monoclonal antibody, the polymer pro-drug nanoparticles can be targeted and enriched in M2-type macrophages. It is proved that this strategy can reduce the probability of drug clearance, promote drug aggregation and release at the tumor site, and improve the anti-tumor effect.” To show targeting to M2-macrophages the authors need to show uptake in a cell line lacking CD163 on the plasma membrane. Figure 6 provides no evidence as to drug clearance, drug aggregation and Dox release following incubation here. Also, how much NP was added here? Methods suggest 6mg/L and this is over 10X the IC50. How healthy are the cells? If brightfield or phase contrast images are available they need to be shown

A molar ratio of 1:30 Ab:linker was used, how did the authors verify that the Ab wasn’t over labelled?

I don’t understand the “CD136 mAb Modified (ug/mg)” is this 2.27ug antibody per 1mg of NP? Can this data be used to estimate the number of antibodies per NP?

I don’t understand what we are looking at in Fig3d, the “state of the nanoparticles” is this a lack of aggregation?

Line 465-6 “macrophages by CCK-8 or MTT assay” which assay was used where? Non-induced RAW cells would be useful here as an additional control, especially if they are shown to have less CD163 on the surface. The viability graphs in Fig 5 may also be clearer as scatter plot, the sigmoidal curves used to calculate IC50 in table 3 can then be fitted to this data

The methods need to state details of the cell lines used, tissue and organism

Please provide centrifuge speeds in x g

Author Response

Response to Reviewers’ Comments on Pharmaceutics-2270735

Dear Reviewer,

On behalf of our authors, I would like to thank you for your careful review. The following shows your comments and our responses.

Reviewer #3:

Comments and Suggestions for Authors

General Comments: In this study Yang et al showed the development of a promising anti-TAM therapeutic from initial nanoparticle development through to in vivo analysis. This is an interesting study that utilises a monoclonal antibody as a targeting factor of their nanoparticle with doxorubicin as the cytotoxic payload. While on the whole this looks like a well performed study. I have some issues with regards to the evidence of targeting and the need for additional controls.

Response: Thank you for your encouraging comments. We have revised the indicated issues accordingly, which have been highlighted in yellow in the revised manuscript and also attached below for your consideration.

Comment 1: The paper utilises an antibody that targets CD163, but there is no confirmation that the cell lines tested have (or lack) CD163. For example, in line 483-4 “which is due to the overexpression of CD163 receptors on the surface of M2-type macrophages” there is no evidence CD163 is overexpressed in the induced RAW264.7 cell line. A Western blot, flow cytometry or immunofluorescence experiment is needed to show relative CD163 levels in RAW264.1 cells before and after induction and the control H22 cells. This is especially important as, as far as I can tell, Proteintech group supply two mAb against CD163 and both are validated against the human version of the protein and both RAW and H22 cells are mouse. (Please also state the product number of the antibody used for the paper).

Response 1: Thank you for your valuable comments. RAW264.7 can be polarized into M2-type macrophages after IL-4 induction, which will upregulate the expression of IL-10 in macrophage, so we used the expression of IL-10 before and after induction to judge the occurrence of polarization. The conclusion that CD163 receptors are overexpressed on the surface of M2-type macrophages has been widely recognized, so we did not measure the expression of CD163 on the cell surface. According to your suggestion, we believe it is necessary to measure the expression of CD163 on the cell surface. Relevant experimental methods have been added in the section of “2.11 Cell Culture”.

Our revisions in the manuscript:

Page 6

“To compare the expression of CD163 on the surface of macrophages before and after induction, we performed the assay by the following method: Unpolarized (RAW264.7) and polarized (M2-type) macrophages were gently pipetted off the dish wall with a pipette, respectively. And the cell membrane was broken by repeated freeze-thawing three times with liquid nitrogen along with the medium. The supernatant was then collected by centrifugation, where CD163 on the cell membrane was soluble in the supernatant, and the CD163 content in the supernatant was determined using the mouse CD163 Elisa kit.”

In addition, we tested the expression of Arginase 1 (Arg1) in cell culture supernatants before and after induction using Elisa kit to further demonstrate that M2-type macrophages were successfully induced. The results of the relevant experiments  are shown below and added to Figure S5.

Thank you for your reminder. The CD163 mouse monoclonal antibody was purchased from Thermo Fisher, Invitrogen, catalog 14-1631-82. The information has been added to the revised manuscript.

Comment 2: Line 500-4 “These phenomena indicate that after the modification of CD163 monoclonal antibody, the polymer prodrug nanoparticles can be targeted and enriched in M2-type macrophages. It is proved that this strategy can reduce the probability of drug clearance, promote drug aggregation and release at the tumor site, and improve the anti-tumor effect.” To show targeting to M2-macrophages the authors need to show uptake in a cell line lacking CD163 on the plasma membrane. Figure 6 provides no evidence as to drug clearance, drug aggregation and Dox release following incubation here. Also, how much NP was added here? Methods suggest 6mg/L and this is over 10X the IC50. How healthy are the cells? If brightfield or phase contrast images are available they need to be shown.

Response 2: Thank you for your valuable comments. The uptake of unmodified mAb-CD163 prodrug nanoparticles (PDNPs) by M2-type macrophages is shown in Figure 6. The results showed that the uptake effect of PDNPs by M2-type macrophages was lower than that of mAb-CD163-PDNPs, indicating the targeting effect of mAb-CD163-PDNPs on M2-type macrophages. From this phenomenon, it can be inferred that it is possible to increase the drug concentration at the tumor site.

In our original manuscript, the results shown in Figure 7e can reflect the experimental evidence regarding drug clearance and drug aggregation. This figure shows a higher concentration of mAb-CD163-PDNPs at the tumor site. But here on lines 515-8, our description seems inappropriate. Therefore, we modify this sentence to "These phenomena indicate that after modification with CD163 monoclonal antibody, the polymeric prodrug nanoparticles are more easily endocytose by M2-type macrophages, which may increase the drug concentration at the tumor site and improve the anti-tumor effect."

In cell uptake experiments, we added NPs based on the drug loading amount. The concentration of DOX in the medium is 6 mg L-1, because if the concentration of DOX is too low, the fluorescence phenomenon is less obvious. At this concentration, the cells were generally healthy because we only co-cultured about 5 h after adding NPs, compared to 48 h in the experiment measuring IC50 values. Unfortunately, brightfield images of cells were not taken at that time, so we recently conduct a similar replicate experiment. In the experiment, we added an equal amount of NPs and co-cultured with M2-type macrophages for 5 h and captured brightfield images of the cells, and the results are shown below. It can be seen that the cells are in good condition.

Comment 3: A molar ratio of 1:30 Ab:linker was used, how did the authors verify that the Ab wasn’t over labelled?

Response 3: Thank you for your comments. Due to the large steric hindrance of the CD163 monoclonal antibodies and the low efficiency of the reaction between -NHS and -NH2 groups, we set the molar ratio of mAb-CD163: NHS-PEG-DBCO to 1:30 for modification. According to the MALDI-TOF MS in Figure 2d, the calculated result is that each CD163 monoclonal antibody was bound to 3.1 DBCO molecules on average, and the Ab was not over-labelled.

Comment 4: I don’t understand the “CD163 mAb Modified (μg/mg)” is this 2.27 mg antibody per 1 mg of NP? Can this data be used to estimate the number of antibodies per NP?

Response 4: Based on the feed ratio and reaction efficiency, we can only calculate the amount of antibody per milligram of nanoparticles as 2.27 μg mg-1. Due to the size distribution of nanoparticles, we are currently unable to accurately calculate the number of antibodies on each nanoparticle.

Comment 5: I don’t understand what we are looking at in Fig 3d, the “state of the nanoparticles” is this a lack of aggregation?

Response 5: Thank you for your comment. Figure 3d is a photo of a nanoparticle solution irradiated with a laser. No significant aggregation of nanoparticles was observed, indicating that the nanoparticles are relatively stable in solution at this time. If a large number of nanoparticles accumulate, bright particles will appear as the laser passes through the solution. To facilitate understanding, we added a sentence in the revised version: "We did not observe nanoparticles aggregation during laser irradiation, indicating that the nanoparticles were relatively stable in solution."

Comment 6: Line 465-6 “macrophages by CCK-8 or MTT assay” which assay was used where? Non-induced RAW cells would be useful here as an additional control, especially if they are shown to have less CD163 on the surface. The viability graphs in Fig 5 may also be clearer as scatter plot, the sigmoidal curves used to calculate IC50 in table 3 can then be fitted to this data.

Response 6: Thanks for your comment. We have modified the sentence to “against both H22 cells by CCK-8 and M2-type macrophages by MTT assay.” Since we have already made antibody-free nanoparticle (PDNPs) pairs as controls, non-induced RAW cells do not need here.

Comment 7: The methods need to state details of the cell lines used, tissue and organism

Please provide centrifuge speeds in x g.

Response 7: Thank you for your comments. We have added the state details about cell lines used in the revised version. The tissues and organs involved in this article are all from balb/c mice. All the centrifuge speeds have been changed to x g.

Our revisions in the manuscript:

Page 6

Human umbilical vein endothelial cells (HUVEC cells), mouse hepatoma cells (H22 cells), and mononuclear macrophageleukemia (RAW264.7 cells).

Best regards,

Peihong Ni (Ph.D.)
Professor in Macromolecular Science
College of Chemistry, Chemical Engineering and Materials Science
Soochow University 
Suzhou 215123
China

Round 2

Reviewer 3 Report

I feel the authors have addressed my concerns and I am happy with their revisions